# Burden, Outcome, and Comorbidities of Extrahepatic Manifestations in Hepatitis C Virus Infection

**DOI:** 10.3390/biology12010023

**Published:** 2022-12-22

**Authors:** Busara Songtanin, Kenneth Nugent

**Affiliations:** Department of Internal Medicine, Texas Tech University Health Sciences Center, Lubbock, TX 79430, USA

**Keywords:** extrahepatic manifestation, hepatitis C virus, hepatitis C infection, antiviral therapy, burden, morbidity, mortality

## Abstract

**Simple Summary:**

Hepatitis C is a liver infection caused by the hepatitis C virus and is a major health problem that contributes to the global burden of chronic disease. Chronic infection can lead to liver cancer and death from end-organ damage. Despite the introduction of novel anti-viral therapy, the disease burden is still high. This review focuses on the various extrahepatic manifestations of the hepatitis C virus, including frequency, mechanism, and outcomes. Despite the introduction of direct-acting antiviral drugs which have more than a 90% rate of sustained virologic responses, about one-third of patients with chronic HCV infections still develop at least one extrahepatic manifestation.

**Abstract:**

Hepatitis C virus (HCV) is a significant cause of chronic liver diseases worldwide and is associated with negative consequences, including cirrhosis, hepatic decompensation, hepatocellular carcinoma, and increased risk of mortality. In addition to liver-related morbidities, HCV is also associated with several extrahepatic manifestations, including mixed cryoglobulinemia, diabetes mellitus, cardiocerebrovascular disease, lymphoma, and autoimmune diseases. These non-liver-related complications of HCV increase the complexity of this disease and can contribute to the economic burden, morbidity, quality of life, and mortality throughout the world. Therefore, understanding how this virus can contribute to each extrahepatic manifestation is worth investigating. Currently, the advancement of HCV treatment with the advent of direct-acting anti-viral agents (DAAs) has led to a high cure rate as a result of sustained virologic response and tremendously reduced the burden of extrahepatic complications. However, HCV-associated extrahepatic manifestations remain a relevant concern, and this review aims to give an updated highlight of the prevalence, risk factors, associated burdens, and treatment options for these conditions.

## 1. Introduction

Over 58 million people worldwide are infected with the hepatitis C virus (HCV), an estimated 2.4 million people in the United States live with hepatitis C, and about 400,000 people died from this disease in 2016 [1,2]. Chronic hepatitis C is associated with significant morbidity, and although the number of cases is decreasing, it is still a common reason for liver transplantation in the United States [3].

Approximately 25% of HCV-infected patients spontaneously clear the infection [4], but most patients become chronically infected with HCV and develop liver-related complications, including decompensated cirrhosis and hepatocellular carcinoma (HCC), which significantly contribute to mortality. However, non-liver-related hepatitis C manifestations can also develop in chronically hepatitis C-infected patients, and these extrahepatic manifestations also contribute to the disease burden, poor outcomes, and mortality in HCV-infected patients [5,6,7].

Extrahepatic manifestations of HCV can involve almost every organ system in the human body and include metabolic syndromes (diabetes mellitus, cardiovascular disease, cerebrovascular disease), autoimmune diseases (Sjogren syndrome, thyroiditis, arthritis), immune-mediated disorders (mixed cryoglobulinemia), malignancy (lymphoma), dermatologic conditions (lichen planus, porphyria cutanea tarda), and renal diseases [8,9,10] (Figure 1). These extrahepatic manifestations of HCV can increase mortality in chronic hepatitis C-infected patients and increase the risk of developing hepatic fibrosis and HCC; they also reduce the quality of life in patients and increase health care costs worldwide.

The primary event in hepatitis C infections involves viral replication in hepatocytes. Chang and co-investigators measured HCV replication in the human liver using in situ hybridization techniques to measure the HCV genome and replicative intermediate ribonucleic acids (RNAs) [11]. They determined that the number of HCV genomes ranged from 7-64 RNA molecules in individual hepatocytes. The maximum number of RNA genomes for a single cell was 74, and the number in the entire liver ranged from 1.8 × 10^11^ molecules to 1.8 × 10^12^ molecules. There was a gradient of dispersion around infected hepatocytes which suggested that infection spread to neighboring hepatocytes as the mechanism of viral spread in the liver. In addition, viral synthetic activities can compromise the normal metabolic activities in hepatocytes and increase the possibility of hepatocellular injury and death. The number of genomes per milliliter (mL) serum in the Chang study ranged from 3.4 × 10^6^ molecules to 5.0 × 10^8^ molecules. Schijman et al. determined the HCV load in 245 male and female patients with HCV infection. The median HCV load was 344,000 international units/mL [12]. There were no major differences between male and female patients or between different viral genotypes (1a, 1b, 2, 3a, 4). These virions in the serum have the potential to reinfect hepatocytes and infect extrahepatic tissues.

The development of extrahepatic complications associated with hepatitis C infection involves complex interactions which include direct viral effects on tissue, the metabolic effects associated with hepatic infection and injury, and the host defense responses associated with ongoing infection [13]. Other factors which potentially influence the development of extrahepatic manifestations include obesity, alcohol use, and the viral genotype causing the infection. Metabolic consequences will also depend on the duration of the infection, the possibility of co-infection with other viral pathogens, and drug treatment effects. These various possibilities are discussed below in the sections on non-hepatic organ involvement in hepatitis C infections.

With new HCV treatments based on pangenotypic direct-acting antiviral (DAAs) therapy, over 90% of hepatitis C infected patients can have sustained virologic responses (SVR) within 2–3 months, and these regimens can be used in many patients with comorbidities who previously could not be treated [14]. Further, recent studies show that SVR was associated with a significant reduction in the risk of several extrahepatic manifestations of HCV [15]. HCV treatment can reduce medical costs by up to $25,000 per patient per year [16]. Therefore, the purpose of this review is to analyze the risk factors, disease burden, outcomes, and comorbidities of each extrahepatic manifestation of HCV to identify possible research priorities for future investigation. Despite the introduction of DAAs and the more than 90% rate of SVR, about 38% of patients with chronic HCV infection develop at least one extrahepatic manifestation [17] (Table 1 and Table 2).

## 2. Diabetes Mellitus, Cardiovascular and Cerebrovascular Diseases

### 2.1. Diabetes Mellitus

Extensive research on metabolic syndromes in HCV patients has been undertaken. Metabolic syndrome (diabetes mellitus, cardiovascular disease, stroke) is highly prevalent among HCV-infected patients. Insulin resistance has been mentioned in HCV-related diabetes mellitus for decades and is associated with poor outcomes and the development of HCC [44,45,46] (Figure 2).

### 2.2. Risk Factors and Prevalence

Several studies have reported that type 2 diabetes (T2DM) is associated with chronic HCV infection [47,48,49,50]. The average pooled prevalence of DM in hepatitis C virus-infected patients is 13–18%, with a higher prevalence in East Asian populations (15.6–22.9%) [9,18]. Further, patients with chronic hepatitis C have three times increased risk of developing T2DM compared to healthy controls and patients with other liver diseases (i.e., chronic hepatitis B infection, cirrhosis) [28,50]. The development of cirrhosis after chronic HCV infection further increases the risk of T2DM (19.6% to 50%) since the associated liver insufficiency also inhibits glucose metabolism [29]. In addition, age older than 40, male gender, family history of DM, and HCV genotype (1,2,4) are risk factors for T2DM in HCV patients [28].

### 2.3. Mechanism

The mechanism underlying diabetes in hepatitis C-infected patients has been extensively studied [51,52]. The most important factors include viral effects on insulin resistance (both intra- and extra-hepatic) and the altered glucose metabolism associated with cirrhosis or fibrosis secondary to HCV infection [53]. Other mechanisms involve pro-inflammatory cytokines, chemokines, and other immune-mediated mechanisms [54].

The hepatitis C virus has acute metabolic effects, which cause insulin resistance in both the liver and extrahepatic tissues and an increase in lipid stores in the liver [55]. Insulin binding to insulin receptors on the cell surface causes autophosphorylation of the cytoplasmic domains of this receptor. This activates insulin receptor substrates 1 and 2 which, in turn, activate downstream enzymatic activity, resulting in the translocation of the glucose transporter 4 (GLUT 4) from intracellular storage sites to the plasma membrane and increases glucose uptake into skeletal muscle tissue and adipose cells [28]. Insulin also inhibits hepatic glucose output by reducing gluconeogenesis. Consequently, insulin resistance both increases hepatic glucose production and reduces uptake by muscle cells. Activation of downstream enzymes by insulin receptor substrates 1 and 2 increases protein and glycogen synthesis, additionally, insulin stimulates triglyceride synthesis and inhibits lipolysis. Viral infection alters hepatic glucose metabolism, decreases glycogen synthesis, and increases glucose levels in the blood. Virus-induced insulin resistance primarily affects extrahepatic organs; hepatic insulin resistance seems to depend on the stage and degree of hepatic damage. HCV viral proteins can increase the autophosphorylation of serine amino acids in insulin receptor substrates 1 and 2 and reduce the phosphorylation of Akt (a serine/threonine kinase that regulates glucose metabolism) [28]. Core viral proteins also increase the production of tumor necrosis factor-alpha (TNF-α) [51]. HCV non-structural protein 3 increases oxidative stress; HCV nonstructural protein 5 promotes lipid accumulation and the formation of reactive oxygen species. Tumor necrosis factor-alpha blocks the phosphorylation of insulin receptor substrates 1 and 2 and reduces GLUT 4 translocation to the plasma membrane [56].

Hepatitis C virus infections also alter lipid synthesis; very low-density lipoproteins (VLDL) are used in the synthesis of viral particles. This alteration in lipid synthesis can cause lipid toxicity which has additional adverse cellular effects, such as apoptosis, the production of reactive oxygen species, and inflammatory/immune responses. In particular, the production of TNF-α has important effects on the liver and other non-hepatic tissue [28,56,57]. Tumor necrosis factor-alpha can increase the serine phosphorylation of insulin receptor substrate 1. Macrophages in adipose tissue produce pro-inflammatory cytokines, such as TNF-α, interleukin-6 (IL-6), and C-reactive protein. Elevated free fatty acids in the circulation stimulate the formation of reactive oxygen species, which can result in cellular injury and inhibit insulin receptor substrate 1 [57].

Steatosis occurs in 40 to 80% of patients with chronic hepatitis C [55]. Cofactors, such as alcohol, obesity, and metabolic syndrome, also contribute to the accumulation of fat in the liver. The hepatitis C virus stimulates de novo lipid synthesis to increase HCV lipoprotein viral particle assembly, promote the synthesis of phospholipids and lipotoxic ceramides, and inhibit mitochondrial fatty acid oxidation. Viral particle assembly interferes with normal VLDL assembly and export. In summary, hepatitis C infections cause insulin resistance and hyperglycemia, along with lipid accumulation resulting in steatosis.

A study by Kawaguchi and colleagues demonstrated that insulin resistance levels were reduced in patients with SVR and remained unchanged in non-responders. In addition, HCV clearance improved beta-cell function and hepatic expression of insulin receptor substrates 1 and 2 [58]. Gao et al. published a meta-analysis that analyzed the association between SVR to pegylated interferon plus ribavirin and insulin resistance [59]. This analysis included 5 studies with 576 chronic hepatitis C patients and found a significant association between SVR and improved insulin resistance measured with the homeostasis model assessment of insulin resistance. Noureddin et al. prospectively studied the effect of SVR using DAAs drugs on liver enzymes and fibrosis scores. They reported a significant decrease in liver enzymes and the ultrasound-based score for liver fibrosis. The study included only 1 measurement of steatosis, and found that some patients with steatosis had an increased fibrosis score following a sustained viral response [60]. Castera and associates studied the effect of antiviral treatment with interferon-alpha on liver steatosis in patients with chronic hepatitis C who had undergone 2 liver biopsies. The patients who had SVR had a significant reduction in steatosis based on histologic grading of the biopsies. Using a multivariate analysis, these investigators found that a decrease in steatosis was independently associated with a SVR during antiviral treatment, severe steatosis on the initial biopsy, HCV genotype 3, and BMI greater than 25 kg/m² (associated with decreased improvement) [61]. Saldarriaga and co-workers compared liver biopsies before and after DAAs treatment for hepatitis C. They used a machine learning algorithm to analyze images from liver biopsies to determine the percent steatosis. All patients in this study had normal liver enzymes at follow-up. However, some biopsies had increased steatosis and fibrosis at follow-up, and 2 patients developed cancer [62]. Overall, these studies suggest that sustained responses to antiviral therapy are associated with reduced insulin resistance and improvement in hepatic fibrosis. However, some patients have sustained steatosis and develop important hepatic complications. Consequently, patients need continued follow-up following drug treatment regardless of their response to treatment.

### 2.4. Burden and Outcome after Treatment

Insulin resistance is an important prognostic factor for morbidity in HCV-infected individuals and is directly associated with resistance to antiviral therapy [63]. In addition, insulin resistance is associated with the progression of hepatic fibrosis and the development of HCC [30,31]. A population-based case-controlled study indicated that diabetes is an independent risk factor for HCC, and the combination of HCV and diabetes was associated with a 37-fold increase in HCC incidence [64]. Several studies have reported that achieving SVR reduces the risk of developing diabetes mellitus in hepatitis C-infected patients [65,66,67] which is likely due to the decreased insulin resistance and improved beta cell function [68]. Hum and colleagues recommend using antiviral therapy as their study showed decreased mean HbA_1_c and decreased insulin use in those who achieved SVR [67]. A recent study by Butt compared HCV patients who received pegylated interferon and ribavirin regimen (PEG/RBV, n = 4764) or a DAAs-containing regimen (n = 21,279) demonstrated that in the DAAs group, the incidence and risk of subsequent diabetes are significantly decreased [69]. Li studied the outcome of 1395 patients with HCV who had diabetes treated with either IFN-based therapy or DAAs; patients with SVR had a significantly decreased risk of developing acute coronary syndrome (HR 0.36), end-stage renal disease (HR 0.46), stroke (HR 0.34), and retinopathy (HR 0.24) [70]. However, patients with DAAs treatment-induced SVR need prolonged follow-up to determine the metabolic complications and outcomes post-treatment.

## 3. Cardiovascular and Cerebrovascular Disease

Atherosclerotic cardiovascular disease is the most common cause of death worldwide [71]. Studies have suggested chronic HCV infection may contribute to the development of cardiovascular disease (CVD) and increase morbidity and mortality rates [19]. In some cases, the viral infection may be overlooked.

### 3.1. Risk Factors and Prevalence

A recent study reported that the risk ratio (RR) of CVD (defined as acute myocardial infarction or stroke) was 1.28 (95% CI 1.15–1.42), indicating that HCV infection is associated with an increased risk of CVD [19]. HCV-infected patients often have significantly lower lipid profiles, including lower low-density lipoprotein (LDL), high-density lipoprotein (HDL), triglycerides, and total cholesterol which may reduce the risk of developing coronary artery diseases [72,73]. Still, HCV infection has been shown to be an independent risk factor for CVD [20,63] and stroke [74,75], but this remains somewhat controversial. Petta and colleagues performed a meta-analysis showing that patients with HCV infection have an increased risk of CVD-related mortality, carotid plaques, and cerebrovascular accident with odds ratios (OR) of 1.65 (95% CI 1.07–2.56), 2.27 (95% CI 1.76–2.94), and 1.30 (95% CI 1.10–1.55), respectively [20]. Conversely, Pothineni and colleagues reported that HCV-infected patients had less obstructive coronary artery disease (CAD) than the non-HCV-infected patient (23% vs. 39%, *p* < 0.05), and that viral load was less likely to be associated with atherosclerosis burden [76]. Another study published by Bilora and colleagues showed a significantly lower rate of carotid atherosclerosis in HCV patients compared to control patients (27% vs. 56%, *p* < 0.005) [77]. The lower lipid levels seen in some HCV-infected patients may reduce the development of atherosclerosis and CAD [78]. The effect of HCV infection on cerebrocardiovascular disease is more pronounced in patients with diabetes or hypertension [20,32]. Further, HCV infection is associated with ischemic electrocardiography findings, cardiac dysfunction, and an increased risk of heart failure [33,34]. In addition, having an HCV/HIV co-infection increases the risk of developing cardiovascular disease compared to those with HIV alone (RR 1.20, 95% CI 1.09–1.32) [19]. Nahon followed 1323 patients with HCV infection with compensated cirrhosis to determine the outcome after achieving SVR. After the introduction of DAAs, the risk of major cardiac events, such as stroke, ischemic heart disease, cardiovascular death, cardiac arrest, and heart failure, was significantly reduced in the SVR group with cirrhosis; viral genotype did not have a role in the risk of cardiovascular events [79]. Studies on HCV infection and the risk of cardiovascular disease are inconsistent, and more studies are needed.

### 3.2. Mechanism

The pathophysiology of cerebro-cardiovascular disease includes both metabolic and non-metabolic pathways. By means of metabolic mechanisms, HCV infection alters glucose and lipid metabolism, resulting in the development of insulin resistance, hepatic steatosis, and T2DM which can lead to the development of atherosclerosis [80]. Non-metabolic pathways include chronic inflammation, endothelial dysfunction, and direct invasion of the arterial wall based on studies finding HCV RNA in human carotid plaques. The pro-atherogenic action of the virus inside the plaque could lead to tissue damage [81,82]. Chronic infection with HCV causes chronic immune stimulation with increased cytokine production (IL-6, TNF-α, C-reactive protein, and fibrinogen) and inflammatory responses which contribute to the development of artherosclersosis [19]. In the Tsui study on heart failure in HCV-infected patients, these patients had a higher incidence of myocarditis and cardiomyopathies and higher levels of TNF-α [83]. Butt demonstrated that at the same level of LDL and total cholesterol, HCV-seropositive males had a higher risk of developing a myocardial infection than males who were HCV-seronegative [84]. However, a systematic review and meta-analysis reported possible bias in exposure-outcome to HCV with the cardiovascular disease since most patients with HCV died from non-cardiovascular diseases [19]. The discussion on insulin resistance in the earlier section on diabetes is relevant to the cardiovascular complications associated with HCV. Patients with HCV infection need frequent follow-ups to diagnose and manage CVD.

### 3.3. Burden and Outcomes after Treatment

Recent studies estimate that around 1.5 million disability-adjusted life years were lost due to HCV-related CVD with a high global burden, especially in low-income countries (South Asian, Eastern European, North African, and Middle Eastern regions) [19]. Further, chronic hepatitis C patients tend to have premature development of CVD [19]. Myocardial perfusion defects were found in 87% of the patients with chronic hepatitis C infection, and this improved with viral eradication using interferon (IFN) therapy [85]. HCV patients had higher mortality rates from CVD with a HR of 1.50 (95%CI 1.1–2.03) [7]. In a cohort study of HCV-infected veterans, SVR with DAAs therapy was associated with a significant decrease in the risk of CVD (HR 0.87, 95% CI 0.77–0.98) [86]. A recent meta-analysis further showed that the risk of stroke also decreases after SVR (HR 0.84, 95% CI 0.74 to 0.94) [87].

## 4. Lymphoproliferative: Mixed Cryoglobulinemia and Lymphoma

HCV replicates in hepatocytes and lymphocytes, and this may cause comorbid lymphoproliferative diseases, including lymphoma and mixed cryoglobulinemia. The association between mixed cryoglobulinemia and hepatitis C infection has been extensively studied [88,89].

### 4.1. Mixed Cryoglobulinemia

#### 4.1.1. Risk Factors and Prevalence

Cryoglobulins are immunoglobulins in the serum that precipitate at temperatures below 37 °C, and cryoglobulinemia refers to the presence of cryoglobulin in a patient’s serum. Clinical manifestations of cryoglobulinemia vary widely in patients, with the most common symptoms being palpable purpura (72% of patients), arthralgias (58%), peripheral neuropathy (21%), and glomerulonephritis (35%) [90]. Specifically, membranoproliferative glomerulonephritis (MPGN) is the most common form of glomerulonephritis, accounting for 80% of cases [91,92,93]. Kidney involvement ranges from isolated proteinuria to renal insufficiency [94]. Studies where HCV-infected patients have 2 times and 17 times increased risk of MPGN and cryoglobulinemia, respectively, compared to non-HCV-infected patients [21]. Another study by Younossi and colleagues showed the pooled prevalence of mixed cryoglobulinemia was 30.1% in HCV-infected patients with an OR of 11.50 (95% CI 4.56–29.00) [9].

#### 4.1.2. Mechanism

Mixed cryoglobulinemia is characterized by a monoclonal expansion of B-cells producing immunoglobulin M (IgM) and is considered a precursor for developing B-cell lymphoproliferative disease. The pathogenesis of mixed cryoglobulinemia probably involves chronic stimulation of lymphocytes by HCV antigens and cytokines [95]. Another possible mechanism is the direct infection of HCV in B lymphocytes which leads to clonal expansion of B cells with IgM and rheumatoid factor release and the potential formation of immune complexes inside the blood vessels, which results in vasculitis [96].

Another study reported an upregulation of B-cell specific cytokines known as BAFF (B lymphocyte stimulator), a critical survival factor for B-cells [97], and increase in transcriptional activity induced by BAFF promoter in HCV-infected patients with mixed cryoglobulinemia compare to HCV-infected patients without mixed cryoglobulinemia [98]. Several studies have reported the induction of multiple cytokines, e.g., IL-6, IL-7, IL-10, by HCV core protein, suggesting a possible contribution to the pathogenesis of mixed cryoglobulinemia and lymphoma [95].

#### 4.1.3. Burden and Outcome after Treatment

Mixed cryoglobulinemia is associated with significantly increased morbidity and mortality, with related renal disease considered the worst prognostic factor [93]. Patients with mixed cryoglobulinemia, in addition to comorbidities such as cardiovascular diseases, liver failure, infections, and chronic renal failure, have an increased mortality risk [35]. HCV-associated cryoglobulinemia vasculitis remains a severe complication with a 5-year mortality rate of 25% [99]. Previously, treatment with IFN-α therapy produced a 40 to 60% SVR with a high prevalence of disease relapse after stopping treatment [100,101,102]. With novel therapy, the introduction of DAAs has led to a higher rate of clinical remission (91%) and SVR in patients with mixed cryoglobulinemia; however, only 70% of patients achieving SVR have a complete clinical response [103]. Approximately 20% of patients achieving SVR have persistent circulating cryoglobulins and reappearance of vascular events, especially in those with severe and/or life-threatening vasculitis. This relapse suggests that the immunologic response has become antigen-independent [104]. Currently, the European Association for the Study of Liver Diseases (EASL) recommends that mixed cryoglobulinemia and renal diseases associated with chronic HCV infection should be treated with pangenotypic DAAs combinations while monitoring for adverse events [105]. El-Serag studied the effect of SVR in 45,260 patients with chronic HCV infection after treatment with DAAs and reported the risk of glomerulonephritis was significantly reduced following SVR [106].

### 4.2. Lymphoma

The close relationship between HCV and lymphoma has been long established, yet the underlying mechanism is still unclear. However, recent studies have helped identify the potential mechanisms underlying this type of lymphoma in these patients.

#### 4.2.1. Risk factors and Prevalence

Chronic HCV infection increases the risk of developing several types of lymphoma, including non-Hodgkin’s lymphoma (NHL), follicular lymphoma, and large B-cell lymphoma. The risk of developing lymphoma in HCV-infected patients is approximately 2.0–2.5-fold higher than in healthy patients [23,95]. The prevalence varies among nationalities with the highest in Egypt (over 20%), then Italy and Japan (5–10%), and then below 5% in most other countries (South Korea, Northern Europe, United States, Australia, and Canada) [23]. The overall risk of developing B-cell NHL in HCV- infected patients who have cryoglobulinemia is as high as 35 times the general population [107]. The most common subtype of lymphoma varies among regions with marginal zone lymphoma appearing more in European and Asian patients [108] and diffuse large B-cell lymphoma appearing more in US patients [36]. The average odd ratios worldwide of marginal zone lymphoma, diffuse large B-cell lymphoma, and lymphoplasmacytic lymphoma are 2.47 (95% CI 1.44–4.23), 2.24 (95% CI 1.68–2.99), 2.57 (95% CI 1.14–5.79), respectively [22].

#### 4.2.2. Mechanism

Information on the mechanism underlying extrahepatic lymphoproliferative diseases is limited. Previous studies have reported an association between HCV and B-cell histologic subtypes. Possible underlying mechanisms include (1) chronic antigen stimulation, (2) increased upregulation of the B-lymphocyte stimulator BAFL, an important factor for supporting B-cell proliferation found in HCV-infected patients [97], (3) direct infection of HCV in B lymphocytes which leads to oncogenic transformation mediated by intracellular virus proteins [95], and (4) DNA damage and mutations in lymphocytes by reactive oxygen species and nitric oxide synthase activated by HCV [109]. One study has shown an increase in BAFF in HCV-infected patients with NHL [87].

#### 4.2.3. Burden and Outcome after Treatment

Previous studies have shown that patients with HCV with follicular lymphoma have a higher risk of developing chronic hepatitis, HCC, and cirrhosis compared to patients with HCV without follicular lymphoma [37]. Several studies have shown that treatment with antiviral therapy causes lymphoma regression in HCV-positive indolent lymphoma patients. Peveling-Oberhag studied the association between SVR (from IFN-based regimens) with lymphoma regression in patients with chronic HCV infection-associated B-cell NHL and found that SVR improved lymphoma response (83% vs. 53%) [110]. A study by Persico shows that DAAs, in combination with chemotherapy, were an independent predictor of disease-free survival by increasing remission of aggressive lymphomas in HCV patients [111]. Currently, the EASL recommends that patients with HCV and lymphoma be treated with pan-genotypic DAAs regimens in combination with specific chemotherapy and careful monitoring of drug-drug interactions [105].

## 5. Renal Diseases

### 5.1. Risk Factors and Prevalence

Data on the prevalence of CKD in patients with HCV are limited. The prevalence of chronic kidney disease and renal insufficiency (serum creatinine ≥1.5 mg/dL) in HCV patients ranges from 5.1% to 17.2% [112]. Renal disease is an important cause of morbidity and mortality in mixed cryoglobulinemia vasculitis induced by HCV [113]. The most common HCV-related renal complication is MPGN or MPGN with type II mixed cryoglobulinemia, with approximately 5% of patients developing renal failure [114]. There is supportive evidence for viral particles infiltrating the renal tissue [115] and for nephrotoxicity secondary to cryoglobulin deposition in the glomerular capillaries and mesangium-forming immune complexes [116]. Clinical manifestations of the cryoglobulinemia-related renal disease range from nephrotic or nephritic renal disease with mild proteinuria to hypertension to renal failure [117]. HCV-infected patients have a 27% increased risk of CKD compared with non-HCV patients (HR 1.27, 95% CI 1.18–1.37) [21]. Other forms of HCV-induced nephropathy without cryoglobulinemia include MPGN without cryoglobulinemia, membranous nephropathy, focal segmental glomerulosclerosis, fibrillary, thrombotic microangiography, and tubulointerstitial injury [118]. In hemodialysis patients, HCV infections are associated with increased all-cause and liver-related mortality.

### 5.2. Mechanism

Hepatitis C virus can cause acute kidney injury in association with cryoglobulinemia-associated vasculitis by direct invasion of HCV into renal parenchyma. HCV RNA has been found in mesangial cells and is associated with a mesangial injury which results in proteinuria [119]. In addition, immune-mediated upregulation of toll-like receptors can occur in the glomeruli in MPGN and cryoglobulinemia, and membranoproliferative disease [120]. Therefore, glomerulonephritis can be classified into two subgroups: cryoglobulinemic and non-cryoglobulinemic immune-complex mediated glomerulonephritis [96]. Cryoglobulinemic glomerulonephritis occurs when cryoglobulins precipitated in the glomerular mesangium interact with complement and initiate a mesangial inflammation [121]. Non-cryoglobulinemic glomerulonephritis occurs without circulating cryoglobulin; this mechanism possibly involves the immune complex of HCV and immunoglobulin G deposition. The glomerular diseases associated with non-cryoglobulinemic glomerulonephritis include membranous nephropathy, IgA nephropathy, and focal segmental sclerosis [96,122].

### 5.3. Treatment

Kidney Disease Improving Global Outcomes (KDIGO) 2018 guidelines recommend that all chronic kidney disease (CKD) patients should be evaluated for proteinuria, hematuria, and eGFR at least annually for detection of HCV-related kidney disease [113]. With DAAs, patients who had SVR have significant improvement in renal function in patients with baseline CKD stage 3–5 [123]. The risk of developing end-stage renal disease (ESRD) is also reduced after DAAs therapy regardless of SVR with an RR of 0.86 (95% CI 0.72, 1.03) [124]. Patients with an estimated glomerular filtration rate (eGFR) < 30 mL/min per 1.73 m^2^ (CKD G4– G5D) should be treated with a ribavirin-free DAAs-based regimen [113]. The EASL recommends that no adjustment DAAs dose is needed when HCV patients have mild to moderate renal impairment (eGFR ≥ 30 mL/min/1.73 m^2^). This Association also recommends the use of grazoprevir and elbasvir in patients with severe renal impairment (eGFR < 30 mL/min/1.73 m^2^), but patients on hemodialysis do not need dose adjustments [105].

## 6. Skin Manifestations in Hepatitis C Virus Infection

A variety of skin diseases are associated with HCV infection; these include mixed cryoglobulinemia, porphyria cutanea tarda (PCT), and lichen planus. Cutaneous manifestations are present in up to 17% of patients with HCV infection [125].

### 6.1. Porphyria Cutanea Tarda

Porphyria cutanea tarda is a dermatological complication that presents with blistering skin lesions forming vesicles or bullae on sun-exposed areas. It is caused by excess uroporphyrin in the skin due to the deficiency of uroporphyrinogen decarboxylase activity, an hepatic enzyme [126,127].

#### 6.1.1. Risk Factors & Prevalence

Multiple studies have demonstrated a strong association between PCT and HCV infection. A systematic review conducted by Gisbert and colleagues analyzed 50 studies and reported the prevalence of HCV in PCT patients was 47% (OR 275, 95% CI 104–725) [24]. Similarly, Younossi and colleagues reported an increased risk of developing PCT in HCV patients (OR 8.53, 95% CI 4.15–17.52) [9]. The prevalence of anti-HCV in PCT varies significantly with geographic distribution; a higher prevalence is found in Southern Europe, and a lower prevalence is found in Northern Europe [128].

#### 6.1.2. Mechanism

The mechanism through which HCV infection may cause or trigger PCT is unknown. One possible pathway involves iron overload and oxidative stress [129]. Chronic infection of HCV can lead to progressive iron accumulation [130]. One study reported that patients with HCV infections with PCT have a higher accumulation of iron in the liver compared to patients with HCV without PCT [130]. Increased hepatic iron has a central role in the pathogenesis of PCT, and factors that increase susceptibility include alcohol, smoking, iron overload, estrogen therapy, homeostatic iron regulator (HFE) mutation, and HCV infection [129,131,132]. Sastre demonstrated that urine porphyrin can be detected in patients with HCV infections who do not have clinical symptoms of PCT and reported the clearance of urine porphyrin after SVR following treatment with DAAs [133].

#### 6.1.3. Burden and Outcome after Treatment

Studies have shown IFN-α treatment may precipitate PCT relapse [134]. Previous studies recommended reducing iron overload by phlebotomy before initiating IFN-based therapies, which produced a better response and improved SVR rates in chronic HCV infection [135]. With DAAs therapy, porphyrin levels are decreased significantly or completely reduced to normal levels, but data are limited [134]. A recent study by García-Fraile recruited 13 patients with HCV infection, and PCT demonstrated that SVR after DAAs treatment leads to PCT resolution [136].

### 6.2. Lichen Planus

Lichen planus is a chronic inflammatory disorder affecting the skin and mucosal surfaces, it is a T-cell mediated disease affecting stratified squamous epithelium of the skin and/or mucus membranes. The classic manifestations include pruritic, polygonal, and purple papules or plaques, and the condition commonly affects middle-aged adults. Lichen planus may appear in the skin, mucous membranes, scalp, nails, and genitalia. Oral lichen planus presents with multiple, symmetrical lesions that can have a reticular, plaque-like, papular, atrophic, erosive, or vesicular-bullous forms in the oral mucus membrane [137].

#### 6.2.1. Risk Factor and Prevalence

The association between HCV and lichen planus remains controversial [138]. A systematic review included 6378 HCV patients with lichen planus and reported the prevalence of HCV infection in lichen planus was 22.3% [139]. Based on a meta-analysis, HCV seropositivity was more prevalent in oral lichen planus patients than controls (OR 6.07, 95% CI 2.73–13.48) [25]. However, a study conducted in central Germany showed no association between lichen planus and HCV [140]. Genetic variability may influence the risk of disease presentation [141].

#### 6.2.2. Mechanism

The exact etiology of lichen planus in chronic HCV infection is still unknown; the pathogenesis may involve an immunological process in which cytotoxic CD8+ T cells cause apoptosis of the basal cells of the oral epithelium [142]. A direct cytopathic effect from HCV in the development of lichen planus is possible [143]. Hepatitis C virus RNA has been found in biopsy specimens. Another hypothesis suggests that circulating autoantibodies in HCV patients promote B-cell proliferation and that host immune response with the production of proinflammatory cytokines as a response to HCV causes skin disease [143]. Genetic factors have been considered a possible factor in the development of oral lichen planus in HCV-infected patients involving HLA-DR6 compared to those without HCV infection. However, this study was conducted in Italy, and geographic differences have been postulated as a factor in developing oral lichen planus [144,145]. Another study showed that patients with oral lichen planus and HCV infection have higher levels of CD8+ lymphocytes in lamina propria compared with patients with oral lichenoid reaction [146]. Figueiredo hypothesized that the host immune system is responsible for oral lichen planus more than direct viral effects [147].

#### 6.2.3. Burden and Outcomes after Treatment

Interferon (IFN) therapy is controversial in the management of HCV in patients with comorbid lichen planus, as there have been reports of both improvement and aggravation of lichen planus symptoms [148,149,150]. Studies on treatment with IFN-free DAAs are limited, and a case series with a small sample reported successful outcomes in HCV-associated oral lichen planus in all seven patients [151].

## 7. Autoimmune Diseases

### 7.1. Sjogren Syndrome

Sjogren’s syndrome (SS) is a chronic systemic autoimmune disease that mainly affects the exocrine glands, such as the lacrimal and salivary glands. The clinical features can be divided into glandular and extra-glandular symptoms, which include dryness of the mouth and eyes, fatigue, and joint pain. This disease predominately occurs in females with a female-to-male ratio of 9:1 [152,153]. Ramos-Casals and colleagues proposed the term “SS secondary to HCV” for subgroups of HCV patients who develop SS [154].

#### 7.1.1. Prevalence and Risk Factors

In a meta-analysis reported by Younossi, the prevalence of SS in HCV patients was 11.9% (95% CI 7.6–16.2) compared to 0.7% (95% CI 0.00–3.3) in non-HCV controls, but this was not statistically different (OR 2.29, 95% CI 0.19–27.09) [9]. Another study from Taiwan showed a strong association between SS and HCV, with the risk of SS being higher in HCV patients (OR 2.49, 95% CI 2.16–2.86) [26]. The clinical symptoms of SS in HCV patients are milder, with fewer systemic manifestations and less evidence of autoimmunity. There is only mild damage to glandular tissues [155]. Anti-SSA or SSB antibodies occurred less frequently in SS-HCV patients than in primary SS patients, but a positive rheumatoid factor (RF) and cryoglobulinemia occurred more frequently [38]. The contributing factors for the development of SS in HCV-infected patients include older age (mean age >55) and liver disease activity. The same study showed no relationship between gender, HCV disease duration, or the presence of cirrhosis and the development of SS-HCV [38].

#### 7.1.2. Mechanism

The pathogenesis of SS-HCV is not completely clear [155]. Several possible mechanisms include tissue injury by proinflammatory cytokines [156] and direct viral infection since several studies have shown that HCV-RNA can be detected in both the serum and saliva [157].

#### 7.1.3. Burden and Outcomes after Treatment

There might be a close relationship between mixed cryoglobulinemia, SS, and B cell lymphoma in chronic HCV infection patients [158]. Patients with SS may develop mixed cryoglobulinemia and may also progress to malignant B-cell non-Hodgkin lymphoma [39]. Of the lymphoma subtypes, MALT lymphoma is the type most frequently found in SS-HCV patients [40]. Only a few studies have analyzed the effect of SVR on SS-HCV patients. One study did reveal that achieving SVR potentially reduces Sjogren’s syndrome [159].

### 7.2. Rheumatoid Arthritis

Arthralgia is the most common rheumatic manifestation and is one of the most common extrahepatic manifestations of HCV infection; it is reported in 40–80% of HCV-infected patients [8,160]. However, true arthritis is less common in HCV-infected patients (approximately 2% to 20%) [161]. The clinical presentation in these patients includes arthralgia, myalgia, mixed cryoglobulinemia, and SS [162]. Many cases are asymptomatic; a study showed that up to 96% of HCV-infected patients have early articular changes, benign disease progression, and absent subcutaneous nodules [163,164]. An important tool used to distinguish rheumatoid arthritis (RA) from HCV-associated RA is the presence of anti-CCP antibodies since HCV-associated RA does not induce anti-CCP antibodies [165].

#### 7.2.1. Prevalence

The prevalence of RA in chronic HCV infection is higher than in normal populations worldwide (OR 2.39, 95% CI 1.52–3.77) and in Asian populations (OR 2.49, 95% CI 1.79–3.45), as reported by Younossi and colleagues [9,18]. Independent risk factors for developing RA in HCV-infected patients include smoking and a previous diagnosis of arthritis [41].

#### 7.2.2. Mechanism

The pathogenesis of RA in patients with chronic HCV infection is unclear, but several mechanisms have been suggested, including direct viral invasion in synovial cells leading to a local inflammatory response [164,166]. Specifically, it is proposed that HCV infection causes immune complex deposition in synovial tissue, and this leads to an inflammatory response. Other potential mechanisms include autoimmune processes, chronic B cell stimulation, and cytokine induction, similar to mechanisms in mixed cryoglobulinemia and lymphoma [164].

#### 7.2.3. Burden and Outcomes of Treatment

The optimal management of HCV-related arthritis has not been established, but expert opinion suggests using non-steroidal anti-inflammatory drugs (NSAIDs), hydroxychloroquine, and low doses of corticosteroid in addition to antiviral therapy to control arthritis-related symptoms [164,167]. The benefit of IFN therapy is still inconclusive as previous studies have shown both clinical improvement and clinical deterioration in HCV-infected patients [164,168]. A national study in Taiwan demonstrated that the risk of RA is reduced after receiving an IFN-based therapy [169]. A study on the impact of SVR on outcomes in chronic HCV infection patients with RA on IFN-based therapies revealed no difference in outcome for those who achieved SVR compared with those who failed to achieve SVR (HR 1.09, 95% CI 0.73–1.64) [15]. A recent study in 65 HCV patients with concomitant RA treated with DAAs demonstrated that these drugs significantly reduce RA activity and improve treatment outcomes with safety and efficacy [170].

### 7.3. Thyroiditis

Thyroid disorders are one of the most common endocrine manifestations in patients with chronic hepatitis C infection, especially thyroid autoimmunity and hypothyroidism [171].

#### 7.3.1. Prevalence and Risk Factors

Autoimmune thyroiditis disease (AITD) and thyroid dysfunction (TD) commonly occur in chronic HCV-infected patients, with a reported prevalence of 20% in patients who are undergoing or have completed IFN-based regimens [172,173]. A study showed the RR of hypothyroidism in HCV-infected patients was 3.10 (95% CI 2.19–4.40) compared to non-HCV-infected patients, and HCV-infected patients tend to have a higher prevalence of anti-thyroid antibodies [27]. A recent study by Nazary showed that patients with chronic HCV infection had a higher prevalence of subclinical hypothyroidism than non-chronic HCV infection patients (6.0% vs. 1.3%; *p*-value: 0.002) [174]. In addition, an increase in the prevalence of anti-HCV antibodies in AITD patients has been reported [42]. AITD occurred more frequently in HCV-infected patients than non-HCV-infected patients and had a higher incidence in females. There was variability in geographic distribution [27,42,43]. There is a higher prevalence of thyroid disorders in HCV-related mixed cryoglobulinemia than in patients with isolated HCV [175,176]. The questions about the association between AITD and HCV include: does AITD develop from IFN therapy or HCV infection? Is HCV an independent risk factor for AITD [177,178,179]? A study from Taiwan showed that female sex (HR 1.49), treatment with PEG-IFN/RBV (HR 1.68), hyperlipidemia (HR 1.38), and history of goiter (HR 6.40) were likely independent risk factors for developing TD in these patients (all *p*-values < 0.001) [180].

#### 7.3.2. Mechanism

The possible pathophysiology of AITD in chronic HCV infection patients may involve the induction of autoimmune responses by IFN to induce an autoimmune phenomenon [181]. Alternatively, HCV may directly infect thyroid cells, resulting in abnormal thyroid function [182].

#### 7.3.3. Burden and Outcomes of Treatment

There are no current studies of IFN-free therapy on the risk of development of AITD and no reports investigating the relationship between SVR and AITD, indicating a need for these types of studies to be conducted. Currently, there are no guidelines for screening AITD in chronic HCV infection patients; however, many studies recommend that clinicians monitor thyroid function with TPO-Ab and TG-Ab in all HCV patients on or off PEG-IFN/RBV therapy, especially females [27,180,183]. A recent study by Wahid evaluated the effect of various DAAs on hypothyroidism and reported an increased risk of developing hypothyroidism in a patient who received Sofosbuvir, IFN, and ribavirin and recommended periodic screening of thyroid function in HCV patients during treatment and post-treatment [184].

## 8. Conclusions

Chronic HCV infection can cause both hepatic and extrahepatic disorders, and both contribute to the morbidity and mortality of this disease. Because patients with hepatitis C infection may have few noticeable symptoms, presentation with extrahepatic manifestations of HCV can help identify hepatitis C in patients. Studies have demonstrated that treatment with DAAs can improve not only hepatic outcomes, but also reduce the symptoms and mortality associated with extrahepatic manifestations. Although many guidelines have recommended treatment for extrahepatic manifestations of HCV (in the form of DAAs and other medications), studies with these disorders remain limited regarding mechanisms specific to HCV-related extrahepatic complications and treatment outcomes in these patients, and therefore continued research is needed in this area. In particular, the pathogenesis and treatment of hepatic steatosis need more study to try to reduce the complications associated with this condition.

## Figures and Tables

**Figure 1 biology-12-00023-f001:**
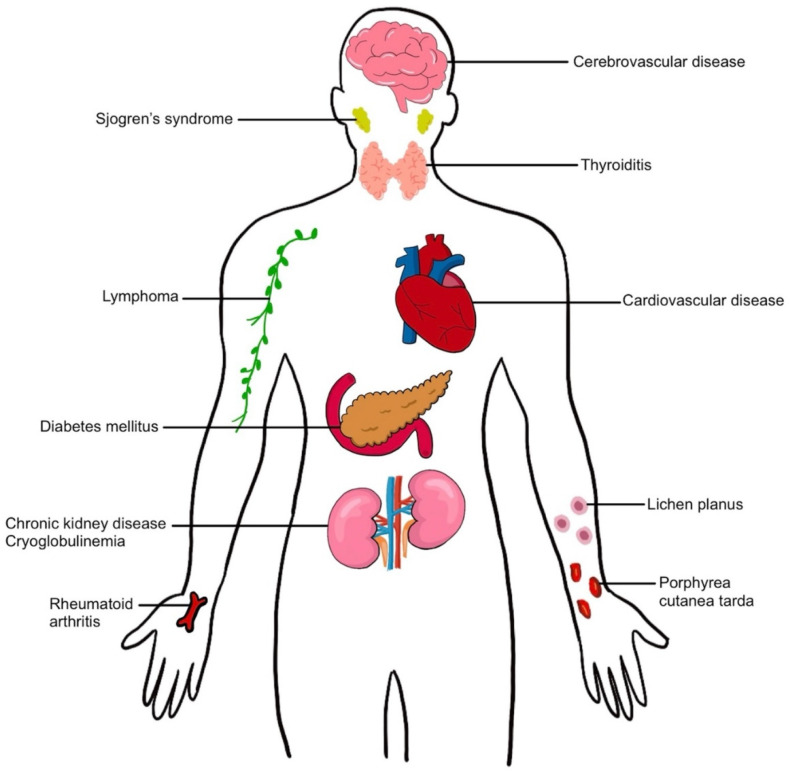
Extrahepatic manifestation in hepatitis C virus infection.

**Figure 2 biology-12-00023-f002:**
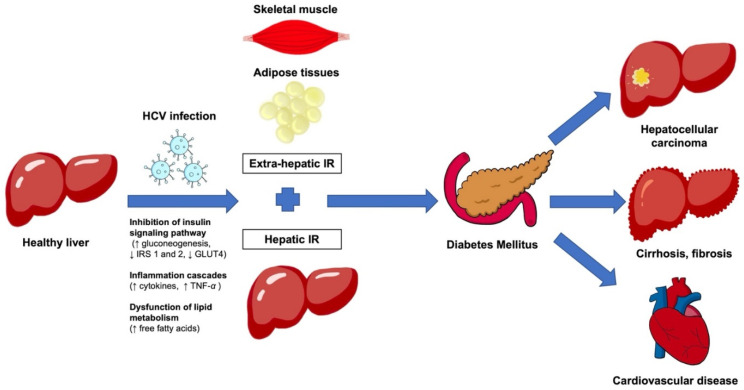
Illustrates the mechanisms associated with HCV infections which lead to hepatic insulin resistance and extrahepatic insulin resistance in skeletal muscles and adipose tissues, which can cause diabetes mellitus and tissue injury associated with hyperglycemia. The chronic inflammation associated with viral infection and insulin resistance can lead to liver fibrosis, cirrhosis, and hepatocellular carcinoma. In addition, some patients develop cardiovascular disease. HCV, hepatitis C virus; IR, insulin resistance.

**Table 1 biology-12-00023-t001:** Prevalence of extrahepatic manifestations in HCV infections.

EHMs	Authors	Study Method	Findings in HCV Patients (95%CI)
**Diabetes** **mellitus**	Younossi [9]	Systematic review (31 studies, n = 263,973)	Prevalence: 15% (13–18%)
	Younossi [18]	Systematic review (21 studies, n = 22,432)	Prevalence 19.0% (15.6–22.9%)
**Cardiovascular and cerebrovascular disease**	Lee [19]	Systematic review (36 studies, n = 341,739)	RR of cardiovascular events, MI, stroke 1.28 (1.15–1.42), 1.13 (1.00–1.28), 1.28 (1.18–1.39), respectively
	Petta [20]	Systematic review (22 studies, n = 390 602)	OR of CVD–related mortality, carotid plaques, and CVA 1.65 (1.07–2.56), 2.27(1.76–2.94), 1.30 (1.10–1.55), respectively
**Mixed cryoglobulinemia**	Younossi [9]	Systematic review (21 studies, n = 4415)	Prevalence: 30% (21.4–38.9%) OR 11.50 (4.56–29.00)
	Park [21]	Retrospective cohort (n = 55,646)	HR 16.91 (12.00–23.81)
**Chronic kidney disease**	Park [21]	Retrospective cohort (n = 56,448)	HR of 1.27 (1.18–1.37)
**Lymphoma**	de Sanjose [22]	Case control (n = 11,053)	OR of Marginal zone lymphoma, DLBCL, and lymphoplasmacytic lymphoma 2.47 (1.44–4.23), 2.24 (1.68–2.99), 2.57 (1.14–5.79), respectively
	Pozzato [23]	Systematic review (50 studies, n = 21,262)	RR of NHL 2.3 (1.8–2.9)
**Porphyria cutanea tarda**	Gisbert [24]	Systematic review (50 studies, n = 2167)	Prevalence: 47–50% OR 275 (104–725)
	Younossi [9]	Systematic review (7 studies, n = 970,315)	Prevalence: 0.5% (0.1–0.8) OR 8.53 (4.15–17.52)
**Lichen planus**	Alaizari [25]	Systematic review (19 studies, n = 4326)	OR 6.07 (2.73–13.48)
**Sjogren syndrome**	Younossi [9]	Systematic review (11 studies, n = 38,789)	Prevalence: 11.9% (7.6–16.2%) RR 2.29 (0.19–27.09)
	Yeh [26]	A population-based analysis (n = 48,145)	OR 2.49 (2.16–2.86)
**Rheumatoid arthritis**	Younossi [9]	Systematic review (4 studies, n = 210,538)	Prevalence: 1% (0.0–2.0%) OR 2.39 (1.52–3.77)
	Younossi [18]	Systematic review (5 studies, n = 18,234)	Prevalence: 4.5% (0.6–25.7%) OR 2.49 (1.79–3.45)
**Thyroiditis**	Shen [27]	Systematic review (12 studies, n = 3603)	Prevalence of hypothyroidism: 6.36% OR 3.10 (2.19–4.40)

EHMs, extrahepatic manifestations; OR, odd ratio; RR, relative risk; HR, hazard ratio; MI, myocardial ischemia; CVD, cardiovascular disease; CVA, cerebrovascular accident; DLBCL, diffuse large B cell lymphoma; NHL, Non-Hodgkin’s lymphoma.

**Table 2 biology-12-00023-t002:** Independent factors and disease burdens of EHMs.

EHM	Independent Factor	Disease Burden
Diabetes mellitus [28,29,30,31]	Cirrhosis, aging, obesity, family history of DM, HCV genotype (1,2,4)	Increased risk of hepatic fibrosis, Increased risk of HCC
Cardiovascular disease [19,20,32,33,34]	DM, HTN, HIV coinfection	Increased risk of MI, cardiac dysfunction, heart failure
Mixed cryoglobulinemia and renal disease [21,35]	Cardiovascular disease, liver failure, infections, chronic renal failure	Increased risk of CKD
Lymphoma [36,37]	Geographic variations	Increased risk of developing chronic hepatitis, cirrhosis, and HCC
Sjogren syndrome [38,39,40]	Older age, liver disease activity	May increase risk of developing MALT lymphoma, malignant B cell non-Hodgkin lymphoma
Rheumatoid arthritis [41]	Smoking, previous history of arthritis	Data limited
Thyroiditis [27,42,43]	Female, geographic variability	Data limited

DM, Diabetes mellitus; HCV, hepatitis C virus; HCC, hepatocellular carcinoma; HTN, hypertension; HIV, human immunodeficiency virus; MI, myocardial ischemia; CKD, chronic kidney disease.

## Data Availability

Not applicable.

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
