# Peer review of "Burden, Outcome, and Comorbidities of Extrahepatic Manifestations in Hepatitis C Virus Infection"

_biology, 2022, doi:10.3390/biology12010023_

Round 1

Reviewer 1 Report

38 - 70mln is not current data - corect it; you can use latest Global Hepatitis Report

42 - "some HCV-infected" - correct it; use detailed data

50-53 - grammar, style of the sentence have to be correct

Author Response

We have changed the data and the grammar and style of the sentence per reviewer suggestions.

Reviewer 2 Report

The present manuscript aims at providing an update on the burden, mechanisms, and risk fcators for extraehepatic manifestation (EHM) of chronic HCV infection. Although the availability of DAA as effective (SVR up to 99% of patients treated) and safe therapy, the topic addressed by the authors is interesting and still clinically relevant.

The major concern of the present manuscript is that appears mostly as a list of possible extrahepatic manifestations without an in dept critical analysis from the authors. In addition, almost al paragraphs on related mechanisms are too general and did not detailly explore the mechanisms responsible for the development of extrahepatic comorbidities. Finally, the distribution of the paragrphs is sometimes confused; paragraph 4 is "Lymphoproliferative: Mixed Cryoglobulineamia and Non-Hoddgkin's Lymphoma", which is followed by "Renal Diesease" and then by "Lymphoma" where the authors mainly report data on NHL.

In view of these considerations, the present manuscript need to be profoundly ameneded before possible consideration by Biology.

Author Response

We have added more detailed discussion about the mechanisms underlying each extrahepatic manifestation; these include immunological mechanisms, cytopathic effects, and molecular mechanisms for the reader to learn about the possible mechanisms responsible for the development of extrahepatic complications.

We have added more discussion on lymphoma, renal disease, mixed cryoglobulinemia. We have changed the title “Lymphoproliferative: mixed cyroglobulinemia and non-Hodgkin’s lymphoma” to “Lymphoproliferative: mixed cryoglobulinemia and lymphoma”. We have moved the renal disease to after lymphoma and mixed cryoglobulinemia section.

Reviewer 3 Report

In this review, the Authors address the extrahepatic burden of HCV infection with regard to mechanisms, prevalence and outcomes after antiviral treatment. The review is nicely written and contents are appropriate, with some major burdens. 

- The review does not really go into the different aspects of the subject (looks more like a report than a review), with particular highlight on mechanisms, which are poorly discussed. Figure 2, for instance, does not picture the mechanisms linking HCV to type 2 diabetes. The mechanisms would represent the best source of novelty for this review, and should adequately reported (e.g. citokynes, metabolic pathways, cell-derived impairments). A similar work has already been discussed, with deep insight into mechanisms (e.g. Negro et al, Gastroenterology 2015).

- The metabolic burden of HCV is potentially the most relevant in the clinical setting (leading to persistent chronic hepatitis after viral eradication) with complex mechanisms that are not been addressed and include adipose tissue dysfunction, lipotoxicity and ER stress oxidative pathways (Leslie et al, J Hepatol 2022)

- Skin disease, cerebrovascular disease and thyroid disorders are less common fields that would increase the interest in this review and should more deeply described. 

- All the cited studies seem to support your considerations; a major focus should be pointed out on controversial findings, to improve the need for appropriate healthcare pathways in clinical setting (e.g. Rodia et al, J Endocrinol Invest 2022 on the thyroid outcomes after DAA)

- A focus on extrahepatic replication of HCV is needed to improve the overall plausibility of the review (Blackard et al, Hepatology 2006). 

Author Response

In this review, the Authors address the extrahepatic burden of HCV infection with regard to mechanisms, prevalence and outcomes after antiviral treatment. The review is nicely written and contents are appropriate, with some major burdens. 

- The review does not really go into the different aspects of the subject (looks more like a report than a review), with particular highlight on mechanisms, which are poorly discussed. Figure 2, for instance, does not picture the mechanisms linking HCV to type 2 diabetes. The mechanisms would represent the best source of novelty for this review, and should adequately reported (e.g. citokynes, metabolic pathways, cell-derived impairments). A similar work has already been discussed, with deep insight into mechanisms (e.g. Negro et al, Gastroenterology 2015).

- The metabolic burden of HCV is potentially the most relevant in the clinical setting (leading to persistent chronic hepatitis after viral eradication) with complex mechanisms that are not been addressed and include adipose tissue dysfunction, lipotoxicity and ER stress oxidative pathways (Leslie et al, J Hepatol 2022)

We have added the details underlying pathophysiology of HCV and type 2 diabetes mellitus. We discussed the inflammatory processes, immunological processes and metabolic pathway linked to insulin resistance and development of cardiovascular disease. We have discussed adipose tissue dysfunction and steatosis, lipotoxicity, and the oxidative stress pathway.

- Skin disease, cerebrovascular disease and thyroid disorders are less common fields that would increase the interest in this review and should more deeply described. 

We have added more detail on skin manifestation of hepatitis C virus infection with the possible mechanism and evidence found on lichen planus and porphyria cutanea tarda.

- All the cited studies seem to support your considerations; a major focus should be pointed out on controversial findings, to improve the need for appropriate healthcare pathways in clinical setting (e.g. Rodia et al, J Endocrinol Invest 2022 on the thyroid outcomes after DAA)

We have added some comments on the management issues required for patients with hepatitis C before and after antiviral treatment.

- A focus on extrahepatic replication of HCV is needed to improve the overall plausibility of the review (Blackard et al, Hepatology 2006). 

We have added to the discussion about hepatic replication of HCV and the levels of viruses in the serum during these infections.

Round 2

Reviewer 3 Report

The review has improved through the amendments. No further comments from my side.